# Modelling and Investigation of Crack Growth for 3D-Printed Acrylonitrile Butadiene Styrene (ABS) with Various Printing Parameters and Ambient Temperatures

**DOI:** 10.3390/polym13213737

**Published:** 2021-10-29

**Authors:** Yousef Lafi A. Alshammari, Feiyang He, Muhammad A. Khan

**Affiliations:** 1School of Water, Energy and Environment, Cranfield University, Cranfield MK43 0AL, UK; yousef-lafi-a.alshammari@cranfield.ac.uk; 2Mechanical Engineering Department, Engineering College, Northern Border University, King Fahad Road, Arar 92341, Saudi Arabia; 3School of Aerospace, Transport and Manufacturing, Cranfield University, Cranfield MK43 0AL, UK; Feiyang.he@cranfield.ac.uk; 4Centre for Life-Cycle Engineering and Management, Cranfield University, Cranfield MK43 0AL, UK

**Keywords:** fused deposition modelling, ABS, thermo-mechanical load, raster orientation, nozzle size, layer thickness, stress intensity factor, fatigue crack growth rate

## Abstract

Three-dimensional (3D) printing is one of the significant industrial manufacturing methods in the modern era. Many materials are used for 3D printing; however, as the most used material in fused deposition modelling (FDM) technology, acrylonitrile butadiene styrene (ABS) offers good mechanical properties. It is perfect for making structures for industrial applications in complex environments. Three-dimensional printing parameters, including building orientation, layers thickness, and nozzle size, critically affect the crack growth in FDM structures under complex loads. Therefore, this paper used the dynamic bending vibration test to investigate their influence on fatigue crack growth (FCG) rate under dynamic loads and the Paris power law constant C and m. The paper proposed an analytical solution to determine the stress intensity factor (SIF) at the crack tip based on the measurement of structural dynamic response. The experimental results show that the lower ambient temperature, as well as increased nozzle size and layer thickness, provide a lower FCG rate. The printing orientation, which is the same as loading, also slows the crack growth. The linear regression between these parameters and Paris Law’s coefficient also proves the same conclusion.

## 1. Introduction

Three-dimensional (3D) printing is one of the additive manufacturing (AM) technology and has been developed over the years. In the past, 3D printing was primarily used for prototyping. However, because of its effective operation, freedom of customization, and cost-effectiveness, 3D printing has been used in many important applications in recent years, such as medical, automotive, aerospace, and biomechanical sectors [1,2,3,4,5].

Moreover, fused deposition modeling (FDM)) is one of the most used 3D printing techniques [6,7]. It is layer-by-layer printing, based on computer-aided design (CAD) and computer-aided manufacturing (CAM) [8]. Moreover, the most common materials used in this method are polymers because of their distinguished properties with low cost and light weight, making them suitable for essential applications, such as aircraft wings and wind blades [9].

These structures can experience fatigue failure due to dynamic loads in a complex thermo-mechanical environment [10,11,12]. Compared with other materials [13,14,15,16,17], the crack propagation during fatigue is highly complicated for FDM polymeric structures because of the significant differences in its microstructure due to various printing parameters. Therefore, the effect of 3D printing parameters on mechanical properties was discussed in much research.

Ismail et al. investigated the effect of raster angle and orientation on the mechanical properties of ABS printed parts using tensile and three-point bending tests [18]. Similar work has been carried out by Ziemian et al., whereby the effect of building orientation on fatigue strength of acrylonitrile butadiene styrene (ABS) parts fabricated by 3D printing was evaluated using the tensile test [19]. Furthermore, Sood et al. improved the compressive strength of the FDM ABS specimen by changing building orientation, raster angle and layer thickness, raster width, and gap [20]. Many other FDM polymeric materials were investigated too. The relationship between building orientations and the fatigue life of polylactic acid (PLA) specimens fabricated by 3D printing were tested under ultimate tensile stress with 50–80% nominal values [21]. Tymrak et al. measured and compared the tensile strength and elastic modulus of FDM ABS and PLA parts printed with different building orientations, layer thickness, and raster gap [22]. In another study, four FDM materials, including polycarbonate (PC), ABS, glycol-modified polyethylene terephthalate (PET-G), and PLA, investigated the effect of layer thickness on the impact strength [23]. Wang et al. analyzed the effect of layer thickness on mechanical properties of polyetheretherketone (PEEK) and its fiber-reinforced parts fabricated by 3D printing. Tensile, flexural and impact tests were performed for glass fiber (GF) and carbon fiber (CF) reinforced materials printed with layer thickness from 0.1 to 0.3 mm [24]. In addition to orientation and layer thickness, several other parameters were also evaluated in previous research. Vicente et al. studied the nozzle size influence on mechanical properties of 3D-printed ABS parts via the tensile test. The selected nozzle sizes, 0.4 mm and 0.8 mm, were compared with other printing parameters [25]. Zhang et al. tested the fracture behaviour of a glass fiber-reinforced polymeric joint by tensile with different temperatures ranging between −35 °C and 60 °C [26].

However, when we critically review the previous studies, it is found that the current research still lacks an investigation about the crack growth in a FDM polymeric structure. No research tested the crack propagation in a FDM structure like many similar works on conventionally manufactured polymers. For conventionally manufactured polymeric structures, a large number of studies tested the fatigue crack growth (FCG) rate considering the ambient temperature and loading frequency [27,28,29]. Luo et al. investigated the effect of temperature on the FCG rate of rubber polymer [27]. Kim and wang calculated the Paris’ power law constant C and m of ABS material at different temperatures and frequencies [28]. Similarly, Kim et al. calculated the Paris power law constant C and m in a commercial-grade ABS under different temperatures ranging from −50 °C to 80 °C [29].

It is found that the above research about conventionally manufactured polymers [27,28,29] investigated the FCG rate with the empirical correlation of Paris’ Law because this famous empirical model [30,31] can represent the crack propagation behavior. Roylance reported that the constants C and m in Paris’ law could be affected by the material type, testing temperature, and loading frequency [32]. So, similarly, the printing parameters of the FDM structure definitely can also affect the crack propagation and affect these two constants. 

Therefore, this paper enhanced our previous research [33], which studied the fatigue life of FDM ABS with different printing parameters and ambient temperatures, to further test and investigate the effect of raster orientation, nozzle size, and layer thickness on crack propagation and Paris’ law for FDM ABS. Unlike the test for conventionally manufactured polymer, the dynamic thermo-mechanical loads are considered during tests to simulate the actual working conditions. Thus, the paper also provides an analytical solution to determine the dynamic stress intensity factor (SIF) used in Paris’ law correlation.

## 2. Materials and Methods

### 2.1. Printing Parameters

Like the previous work [33], the paper focused on three essential parameters: building orientation, layer thickness, and nozzle size. These parameters were tested under three different environment temperatures (50, 60, 70 °C). 

Three values, as shown in Table 1 and Figure 1, Figure 2 and Figure 3, were evaluated for each printing parameter to cover the typical range of printing parameters. It was assisted in a complete assessment of the experimental results. Moreover, to deliver excellent print quality and a typical setting range, the layer thickness and nozzle size were derived from the profile of the default setting [33].

### 2.2. FDM Material

The red ABS (Color code: RAL 3020) filament fabricated by the Ultimaker^®^ (Utrecht, The Netherlands) was selected as raw material (Filament details shown in Table 2) [33] because the ABS is the most used material for FDM and plastic zones can be clearly observed for red color material [33].

### 2.3. Sample Preparation

The bending fatigue test of FDM polymers does not have a specific standard. Therefore, the geometry of the sample was selected as 150 × 10 × 3 mm^3^, as shown in Figure 4, which is the same as that used in previous research [33,34]. Therefore, the results for the present experiment and previous research can be compared. There was a 0.5-mm-deep initial-seeded crack near the fixed end of the beam for all the samples to ensure that the maximum stress concentration took place at the same point. As a result, the crack location in all the experiments was the same and very similar to the fatigue failures of a real scenario due to the stress concentration.

The CATIA v5 CAD software was used to design the specimen. The CAD model of the specimen was built in CATIA v5 with an STL file format and imported to the Ultimaker^®^ CURA 4.6 software. CURA software was used to set a series of printing parameters. Most parameters were maintained or recommended to default values during the printing process, apart from the selected parameters shown in Table 1. The printing process of the Ultimaker^®^ 2+ 3D printer is shown in Figure 5.

### 2.4. Experiments

#### 2.4.1. Experimental Scheme

The 81 configurations mentioned in Section 2.1 were tested by a series of experiments. The experiment was divided into three main parts. Firstly, the continuous bending vibration was applied to the specimen. The number of cycles between different crack propagation until fracture were recorded. Secondly, a digital microscope was used to capture the crack depth. Finally, a dynamic mechanical analysis test (DMA) was applied to the broken samples from the previous test to find the storage modulus. After that, the experimental outputs were used in analytical calculations to find out the SIF and FCG rate. The detailed experimental scheme can be shown in Figure 6.

#### 2.4.2. Experimental Setup and Procedures

The experimental setup is shown in Figure 7, which is the same as the previous study [33]. The fundamental frequency of the specimen was measured by an impact test twice. Then, the shaker excited the specimen with an amplitude of 2 mm with the measured fundamental frequency. This led to the resonance of the beam, and then the pre-seeded crack growth started. The accelerometer measured the acceleration and time data during crack growth and transferred them into SignalExpress software via a DAQ card. The test was paused when a significant displacement amplitude drop was observed in Signal Express. A Dino–Lite digital microscope (AnMo Electronics Corporation, Hsinchu, China) measured the corresponding crack depth. The new fundamental frequency was measured and applied to the cracked specimen again. These procedures were repeated until the beam broke.

## 3. Analytical Model and Experimental Data Process

### 3.1. Calculation of SIF Range and FCG Rate

This section shows how SIF and the FCG rates were calculated based on the raw experimental data. Because the dynamic loads were applied on the tests, the stress ratio is time-dependent rather than a constant value. The appropriate estimation is essential to calculate the SIF.

For mode I fracture, the range of SIF (ΔK) can be calculated using Equations (1) and (2) [35,36]. However, the average applied stress range (Δσ) was then used in this paper. It was different from the constant stress range in the standard FCG test. Δσ was time-dependent due to the variable beam amplitude. Therefore, the mean Δσ was approximately calculated using the MATLAB code and then substituted in Equation (1).
(1)$$\Delta K=\Delta \sigma \sqrt{\pi a_{i}}f\left(\frac{a_{i}}{H}\right)$$
where $a_{i}$ the initial crack depth, H is the beam thickness, and $f\left(\frac{a_{i}}{H}\right)$ is the dimensionless boundary correction factor which can be calculated using Equation (2) [35,37].
(2)$$\left(\frac{a}{H}\right)=1.13-1.374\left(\frac{a}{H}\right)+5.749{\left(\frac{a}{H}\right)}^{2}-4.464{\left(\frac{a}{H}\right)}^{3}$$

Moreover, to calculate the average stress range at the crack location according to the accelerometer measurements, the mean stress amplitude was calculated for each loading cycle by using Equation (3) to Equation (12) [38]. Firstly, the displacement amplitude was calculated at the beam tip (accelerometer location) by its quadratic integral relationship with acceleration amplitude measured by the accelerometer during the experiments, as shown in Equation (3).
(3)$$y_{i}\left(L\right)=\frac{1}{2}\frac{acc_{i,peak}-acc_{i,trough}}{{\left(2\pi f_{i}\right)}^{2}}$$
where $y_{i}\left(L\right)$: the displacement amplitude at beam tip in the *i*th cycle, $acc_{i}$: the acceleration in the *i*th cycle, and $f_{i}$: the fundamental frequency of beam in the *i*th cycle.

Then, the displacement amplitude at the crack location $y_{i}\left(l_{c}\right)$ can be derived based on the mode shape of a cracked beam f(x) [39], as shown in Equations (4)–(9)
(4)$$y_{i}\left(l_{c}\right)=C_{sf}\times f\left(l_{c}\right)$$
(5)$$C_{sf}=\frac{y_{i}\left(L\right)}{f\left(L\right)}$$
(6)$$f\left(x\right)=C_{1}\sin\left(\beta x\right)+C_{2}\cos\left(\beta x\right)+C_{3}\sinh\left(\beta x\right)+C_{4}\cosh\left(\beta x\right)$$
(7)$$\beta^{4}=\frac{\omega^{2}\rho A}{E_{T}I}$$
(8)$$I=\frac{bH^{3}}{12}$$
(9)$$\omega =\frac{t_{i}-t_{j}}{i-j}$$
where $C_{sf}$: the scale factor, $l_{c}$: the crack location, $C_{1-4}$: the coefficient in mode shape function of the cracked beam, ω: the fundamental angular frequency, ρ: the FDM ABS density, A: the beam’s cross-section area, $E_{T}$: the elastic modulus at test temperature, I: the area moment of inertia, b: the beam width, H: the beam height; and $t_{i}$ and $t_{j}$: the *i*th and *j*th peak time, respectively.

After that, the bending moment $M_{i}\left(l_{c}\right)$ at the crack location in *i*th cycle can be calculated by Equation (10):(10)$$M_{i}\left(l_{c}\right)=\left|E_{T}I\frac{d^{2}y_{i}\left(l_{c}\right)}{dx^{2}}\right|$$

Next, assuming that the bending stress amplitude $\sigma_{i}\left(l_{c}\right)$ at the crack tip in the *i*th cycle was constant and equal to the stress at the beam surface. It can be calculated using Equation (11),
(11)$$\sigma_{i}\left(l_{c}\right)=\frac{6M_{i}\left(l_{c}\right)}{bH^{2}}$$

Finally, the mean stress range is given by Equation (12)
(12)$$\Delta \sigma =\frac{\sum_{i=1}^{n}\sigma_{i}\left(l_{c}\right)T_{i}}{\sum_{i=1}^{n}T_{i}}$$
where $T_{i}$: the period of the *i*th cycle.

During the dynamic fatigue crack test, the single side cracked beam vibrated up and down. As a result, it was exposed to two types of cyclic loading (tensile and compressive stresses), as shown in Figure 8. However, only tensile stress was responsible for the propagation of the crack. Therefore, the stress amplitude $\sigma_{i}\left(l_{c}\right)$ was used to show the stress range per cycle instead of the difference among the peak and trough as shown in Equation (12).

For the same reason, the actual number of cycles that lead to crack propagation is half of the total number of cycles. Because the crack was not propagated when the specimen was exposed to compressive stresses; therefore, the average crack rate between two crack depths can be calculated by Equation (13)
(13)$$\frac{da}{dN}=2\left(\frac{a_{f}-a_{i}}{N}\right)$$
where $a_{f}$: the final crack depth.

With the calculated SIF range ΔK and FCG rate $\frac{da}{dN}$. The empirical relationship between the ΔK and the fatigue crack growth rate can be modelled. The format of the model was represented by Paris law, as shown in Equation (14) [29,30]. After that, this relationship was plotted, and then the suitable values for C and m were identified using the curve fitting function in MATLAB.
(14)$$\frac{da}{dN}=C{\left(\Delta K\right)}^{m}$$

### 3.2. Data Process

The C and m values were determined in Section 3.1 for different configurations. In order to investigate and compare the effect of printing parameters on Paris’ law, a multiple linear regression (MLR) model was used, as shown in Equation (15).
(15)$$\hat{Y}=b_{0}+b_{1}X_{1}+b_{2}X_{2}+b_{3}X_{3}+b_{4}X_{4}$$
where $\hat{Y}$ is the C or m values, and we denote that $X_{1}$ represents the raster orientation, $X_{2}$ represents the nozzle size, $X_{3}$ represents the layer thickness, $X_{4}$ represents the temperature, and $b_{0-4}$ is the estimated regression coefficient that quantifies the association between the parameters X and the dependent variable $\hat{Y}$.

The printing parameters are converted from the original values in Table 1 into standardized dimensionless values to eliminate the effects of differences in properties, such as dimension and order of magnitude between different variables, thus making the effect sizes of different variables comparable. The z-score standardized method was used, and the values of X are shown in Table 3.

## 4. Results and Discussion

The influence of printing parameters (building orientation, layers thickness, and nozzle size) under 50, 60, and 70 °C on the FCG rate were studied. The effect was visualised by the log-log plot, which used the Paris’ law relationship between the crack growth rate ($\frac{da}{dN}$) during crack propagation and the range of the stress intensity factor (Δ*K*). The first-order polynomial curve fitting method was used to plot the figures due to the linear relationships of Paris law. R-squared and RMSE values for each parameter are shown in Table 4.

### 4.1. Building Orientation Influence

Figure 9 shows the crack growth rate variation for X, XY, and Y building orientations. As can be observed, samples printed with X building orientation have the lowest FCG rate, while Y building orientation has the highest, and XY orientation lies between them for the same SIF value.

Table 5 shows the FCG rate range and the mean fatigue life. The average FCG rate for the X orientation corresponds to the 9.68 × 10^−7^ m/cycle. In comparison, Y orientation corresponds to the 9.77 × 10^−7^ m/cycle. Furthermore, the mean number of cycles until the fracture has 4343 cycles for the X orientation. However, the Y building orientation approximately has 2282 cycles. These results are similar to the previous study provided by the applied bending fatigue test [32,40]. However, they are different from what was provided in most previous studies, which was tested by tension fatigue test [20,41]. This found that the XY building orientation had the highest fatigue life, which means they had the lowest FCG rate. Therefore, the difference in the type of the test may be the main reason behind the difference in results.

The findings are reasonable. The Y building orientation samples provide the highest FCG rate because the Y orientation is lateral on the beam in a similar direction to the initial seed crack. In fracture mechanics, the micro-cracks on or in the structure are the main reason for the propagation of the crack [42]. These micro-cracks occur between filaments of 3D structures due to printing defects in the form of microvoids. Furthermore, one of the stress characteristics is that it concentrates around the weakest area of the structure. Therefore, when the beam is vibrated, these microvoids lead to concentrate the stress around them. The micro air voids are in the same direction as the initial seed crack in the Y building orientation. Therefore, an excellent crack path was created, especially when the stress acting vertically on these voids due to the beam’s vibration increased the FCG rate and decreased the fatigue life. In contrast, the opposite happened in X building orientation [33].

### 4.2. Layer Thickness Influence

Figure 10 shows that the 0.15-mm layer thickness had the lowest crack growth rate, while the 0.05-mm layer thickness had the highest. Furthermore, the 0.10-mm layer thickness always lies between them. Table 6 shows the FCG rate range and the mean fatigue life. The average crack growth rate for 0.15-mm layer thickness corresponds to the 9.19 × 10^−7^ m/cycle. While 0.05-mm layer thickness corresponds to a 1.04 × 10^−6^ m/cycle, and the 0.10-mm layer thickness lies between them. Furthermore, the highest mean number of cycles until the fracture has 3805 cycles for the 0.15-mm layer thickness, while the lowest has 3417 cycles provided by the 0.05-mm layer thickness.

As can be observed, when the layer thickness changes between 0.05, 0.10, and 0.15 mm, the crack growth rate changes slightly. In addition, by increasing the layer thickness, the FCG rate will decrease. These results are similar to what was founded in previous studies for FDM ABS and PLA [33,40,43,44]. 

The reason behind the lower FCG rate of 0.15-mm layer thickness is related to the micro air voids theory which states that, when the micro air voids decrease, the FCG rate will decrease. In addition, when the layer thickness increases, the number of air voids will decrease, increasing the global density and decreasing the concentrated stress on the sample, increasing the strength and fatigue life. Therefore, the 0.15-mm layer thickness has fewer micro air voids (dark red area) than the 0.10-mm and 0.05-mm layer thickness, which provides more strength and longer fatigue life, as shown in Figure 11 [33].

### 4.3. Nozzle Size Influence

Figure 12 shows that the 0.8-mm nozzle size had the lowest crack growth rate, while the 0.4-mm nozzle size had the highest and 0.6 lies between them. However, some samples operating at 60 and 70 °C shows the 0.8-mm nozzle size had the lowest FCG at the start of the test and the highest at the end. Table 7 shows the average FCG rate range and mean the number of cycles until fracture. The average crack growth rate range for the 0.8-mm nozzle size corresponds to the 8.19 × 10^−7^ m/cycle. However, a 0.4-mm nozzle size corresponds with the 1.06 × 10^−6^ m/cycle. Furthermore, the highest mean number of cycles until the fracture has 4147 cycles by the 0.8-mm nozzle size, while the lowest has 3224 cycles by the 0.4-mm nozzle size.

As can be observed, when the nozzle size increases, the FCG rate will decrease. These results are like previous research conclusions for FDM ABS and PLA [33,40,43,44]. Furthermore, this result is related to the micro air voids provided and explained in Section 4.2. The samples printed with 0.8-mm nozzle size has a lower number of micro air void than 0.6 and 0.4 mm, as shown in Figure 13.

### 4.4. Environmental Temperature Influence

The environmental temperature has a significant influence on the FCG rate. Figure 14 shows that the FCG rate increased due to the increase in temperature. Additionally, the 50 °C environmental temperature had the lowest crack growth rate, the 70 °C had the highest, and 60 °C was lies between them. Table 8 shows the average FCG rate and the mean number of cycles. The average crack growth rate range for the 50 °C environmental temperature was the 9.77 × 10^−7^ m/cycle. While the 70 °C environmental temperature corresponds to the 9.86 × 10^−7^ m/cycle. Furthermore, the mean number of cycles until the fracture has 3604 cycles for the 50 °C environmental temperature, 3068 cycles for 60 °C, and 3122 cycles for 70 °C. This means that the FCG rate increases when the temperature increases. This result is likely what was mentioned in previous research [9].

The results are reasonable. The increase in temperature is negatively affecting the mechanical properties. The evidence for this claim is that, in the DMA test, when temperature increases, the storage modulus decreases, decreasing the strength of the printed sample and increasing the FCG rate. In addition, for the FDM ABS at lower temperatures, the strength of the bounding of the molecules to each other is very strong. However, at higher temperatures, this strength decreases, which leads to the slip of the chain easily. In conclusion, the increase in temperature in FDM ABS negatively affects the mechanical and microstructural properties, leading to an increase in FCG rate [33].

### 4.5. Paris Law Constants (C and m) for Different Printing Parameters

The C and m values were determined for different configurations experimentally. Part of the results is shown in Table 9. The full results are attached in Appendix A. It was found that the values in Table 9 for FDM ABS are higher than previous research testing on conventionally manufactured ABS [28,29] because of the micro air voids in the FDM structure. The existence of air voids changes the microstructure of ABS, which leads to lower crack resistance. This is reflected as the larger values of C and m in Paris’ law.

Because there is too much data, the MLR results from Section 3.2 were used to investigate printing parameters’ effect on Paris’ law.

Regression Equations (16) and (17) were fitted by MATLAB.
(16)$$C=0.38+0.55X_{1}-0.45X_{2}-0.57X_{3}+0.02X_{4}$$
(17)$$m=0.66+0.12X_{1}+0.09X_{2}-0.2X_{3}+0.11X_{4}$$

The regression coefficients of building orientation and temperature are positive values (0.55 and 0.02). This proves that, when building orientation tends to be perpendicular to the direction of stress or increased ambient environment, it increases the C value, resulting in a higher FCG rate. In contrast, the nozzle size and layer thickness regression coefficients are negative values (−0.45 and −0.57). This shows that the increased nozzle size and layer thickness all decrease the FCG rate. These results again support the conclusion from Section 4.1, Section 4.2, Section 4.3 and Section 4.4.

The m value in Paris Law, which shows the slope of the curve in log-log plots, has a smaller effect on the FCG rate. Therefore, the regression coefficient of nozzle size for the m value is 0.09, which is the opposite of −0.45 in Equation (16). The overall trend of FCG rate still decreases when the nozzle size is increasing.

## 5. Conclusions

An experimental study was performed on FDM ABS using the dynamic mechanical loads at different ambient temperatures to investigate the influence of printing parameters and temperature on FCG rates and the Paris law constants. 

The paper proposed one analytical solution to determine the SIF range when dynamic stress at the crack location is difficult to measure under structural resonating conditions. The empirical Paris law model was developed for different configurations of raster orientation, nozzle size, layer thickness, and ambient temperature. The corresponding C and m constants are listed as a reference for future research.

With the comparison of the FCG rate, it is found that increased layer thickness and nozzle size all reduce the FCG rate due to the decrease in micro air voids. On the contrary, the building orientation close to 90° and increased temperature accelerates the crack growth. Therefore, combining the following parameters provides the lowest FCG rate and Paris law constant: X building orientation, 0.15-mm layer thickness, and 0.8-mm nozzle size.

For Paris’ law, C and m values were also affected by environmental temperature and printing parameters. As the temperature increased, the C and m values increased. In addition, the X building orientation, higher layer thicknesses, and larger nozzle size values provide the lowest C and m values because C and m values are directly related to FCG rate. These results are realistic due to the material type and environmental temperature influence on crack growth [9]. 

However, the overall trend in C and m values was higher than in previous research [28,29]. This disparity is likely due to differing material status, structure, and operating conditions, leading to a decrease in the FCG rate. 

## Figures and Tables

**Figure 1 polymers-13-03737-f001:**
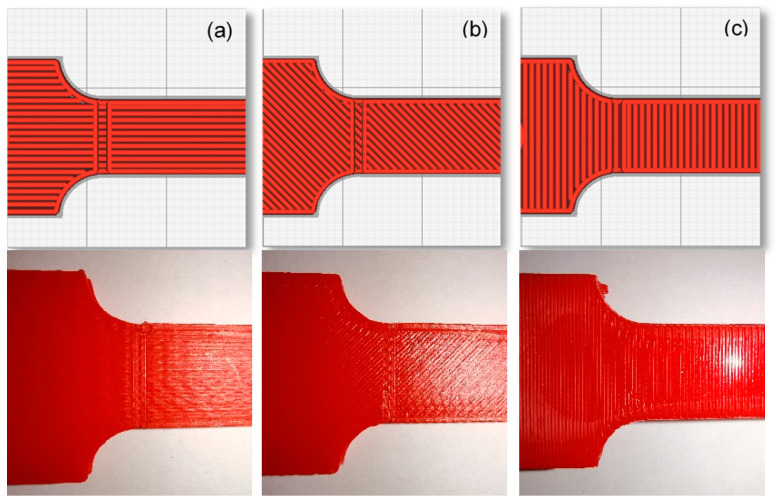
Building orientations directions and angles: (**a**) X (0°); (**b**) XY (±45°); and (**c**) Y (90°) [33].

**Figure 2 polymers-13-03737-f002:**
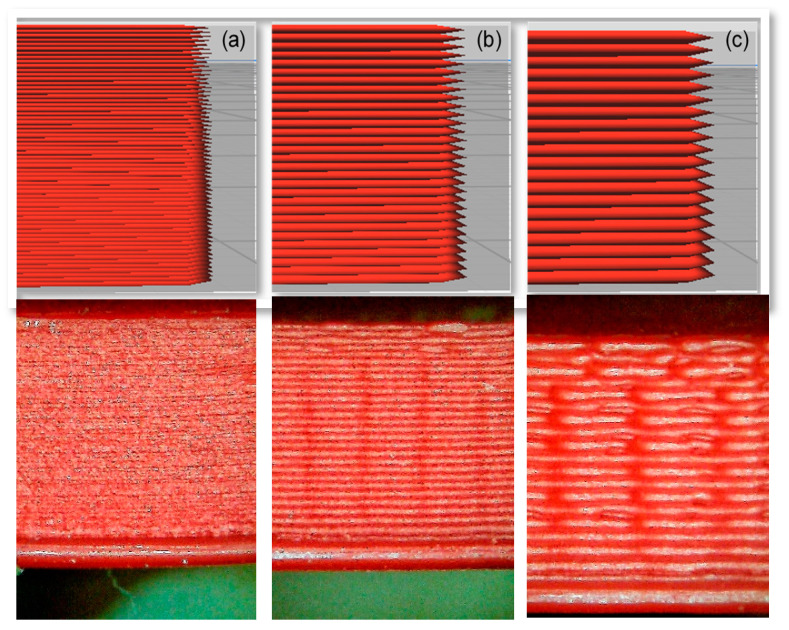
Three different layer thicknesses: (**a**) 0.05 mm; (**b**) 0.10 mm; (**c**) 0.15 mm [33].

**Figure 3 polymers-13-03737-f003:**
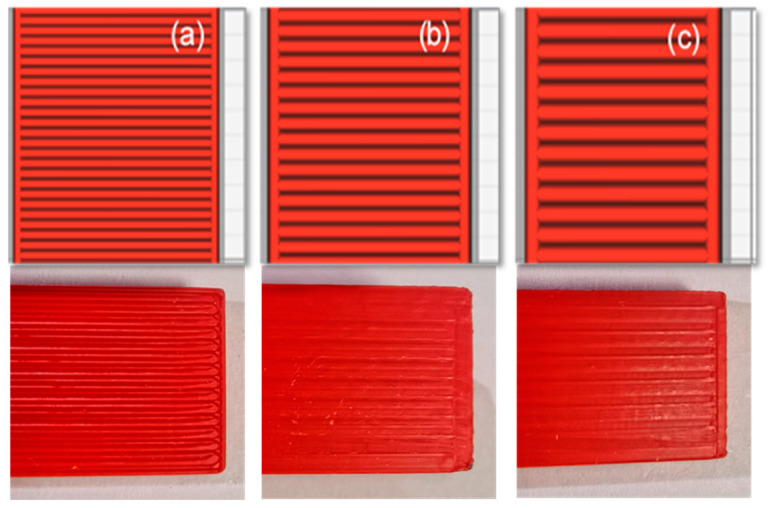
Three different nozzle sizes: (**a**) 0.4 mm; (**b**) 0.6 mm; (**c**) 0.8 mm [33].

**Figure 4 polymers-13-03737-f004:**
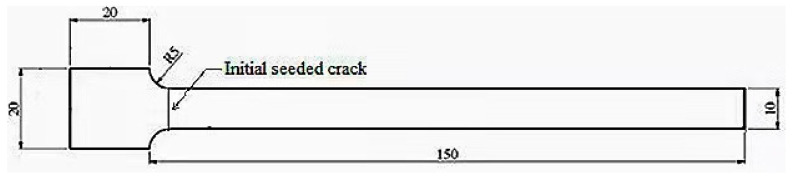
The samples geometry [33].

**Figure 5 polymers-13-03737-f005:**
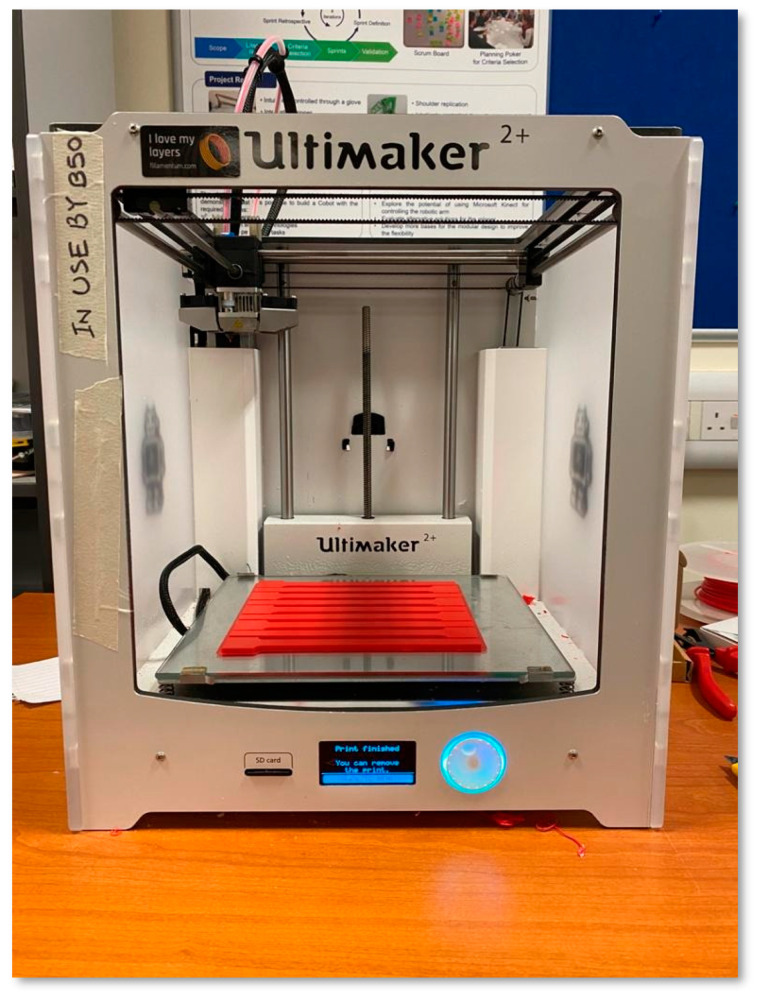
The 3D printing process by the Ultimaker 2+ printer.

**Figure 6 polymers-13-03737-f006:**
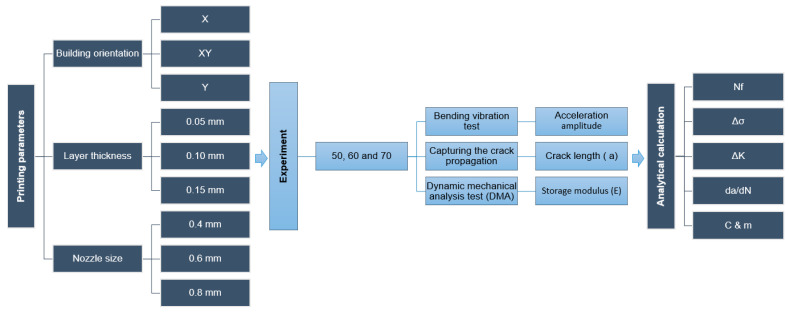
Detailed experiment scheme.

**Figure 7 polymers-13-03737-f007:**
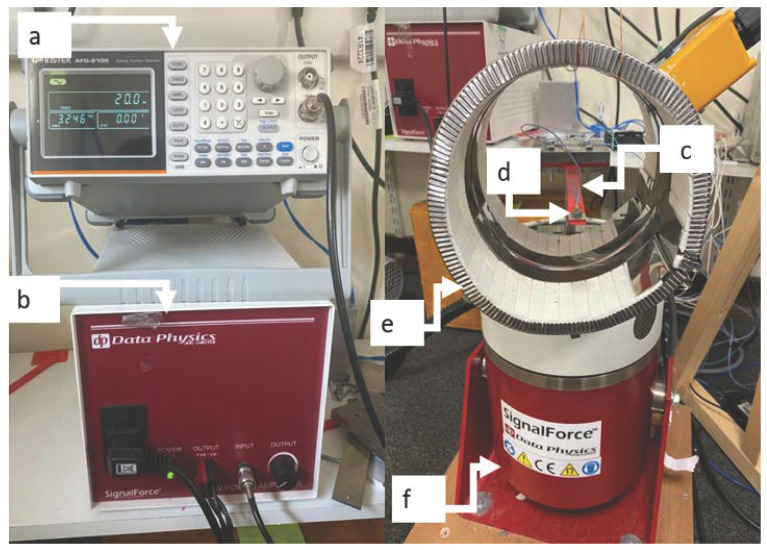
Experimental setup for the bending vibration test. (**a**) Signal generator, (**b**) power amplifier, (**c**) tested sample, (**d**) accelerator, (**e**) heat band, and (**f**) shaker.

**Figure 8 polymers-13-03737-f008:**
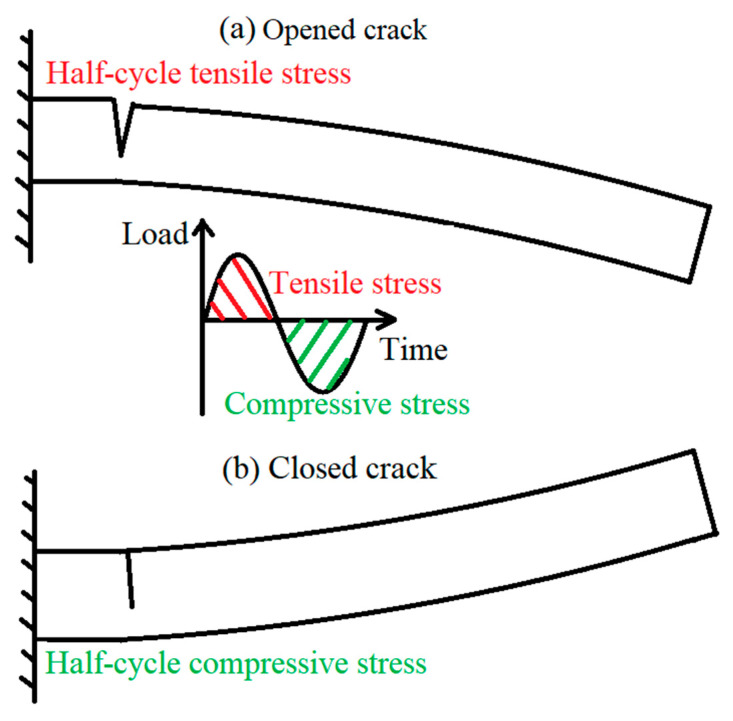
Load conditions during cyclic vibration.

**Figure 9 polymers-13-03737-f009:**
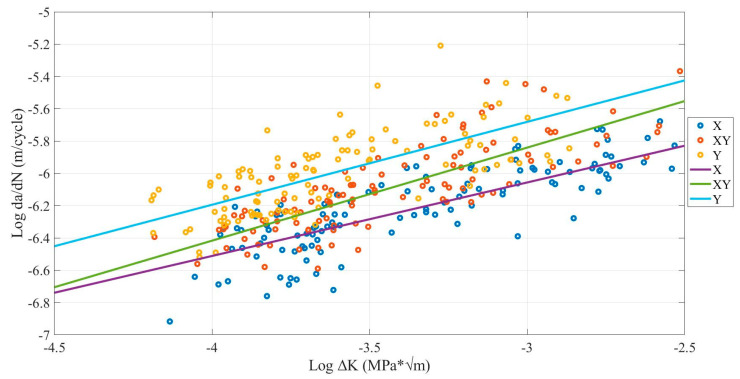
Comparison of building orientation influence on FCG rate for ABS samples with (0.4-, 0.6-, and 0.8-mm nozzle size and 0.05-, 0.10-, 0.15-mm layer thickness) under 50, 60, and 70 °C environmental temperatures.

**Figure 10 polymers-13-03737-f010:**
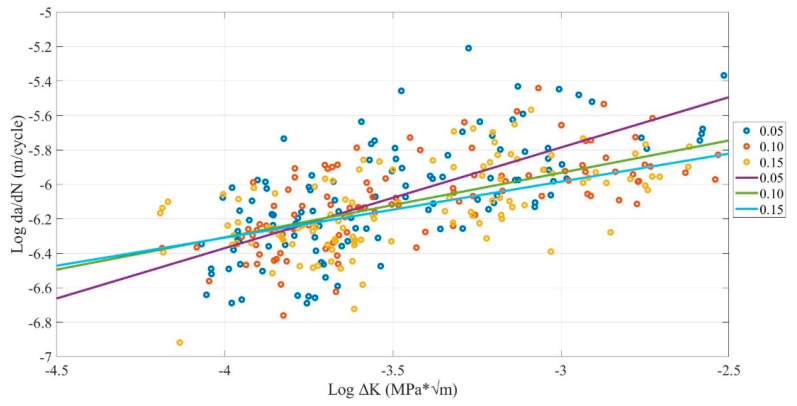
Comparison between layer thickness influence on FCG rate of FDM ABS samples with (0.4-, 0.6-, and 0.8-mm nozzle size, and X, XY, and Y building orientation) under 50, 60, 70 °C environmental temperatures.

**Figure 11 polymers-13-03737-f011:**
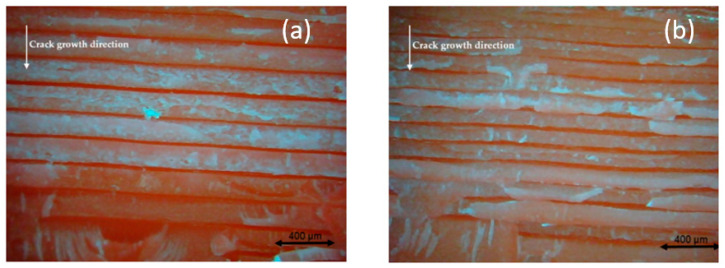
Comparison between the influence of different layer thicknesses of FDM ABS, printed with a 0.6-mm nozzle size, Y building orientation, and layer thickness (**a**) 0.15 mm and (**b**) 0.10 mm [33].

**Figure 12 polymers-13-03737-f012:**
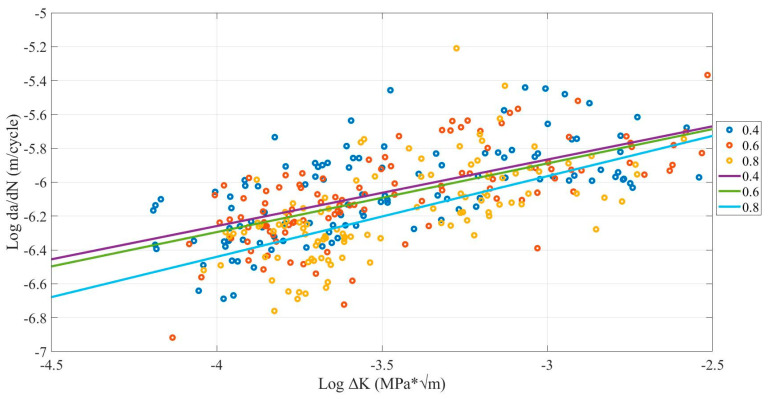
Comparison between nozzle size influence on FCG rate of FDM ABS samples with (0.05, 0.10 and 0.15 mm layer thickness and X, XY, Y building orientation) under 50, 60, 70 °C environmental temperatures.

**Figure 13 polymers-13-03737-f013:**
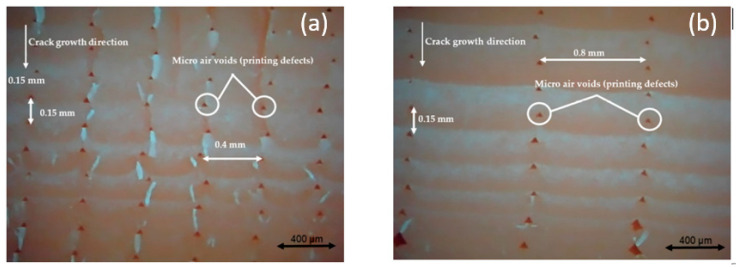
Comparison between the influence of different nozzle sizes of FDM ABS, printed with 0.15-mm layer thickness, X building orientation, and nozzle size (**a**) 0.4 mm and (**b**) 0.8 mm [33].

**Figure 14 polymers-13-03737-f014:**
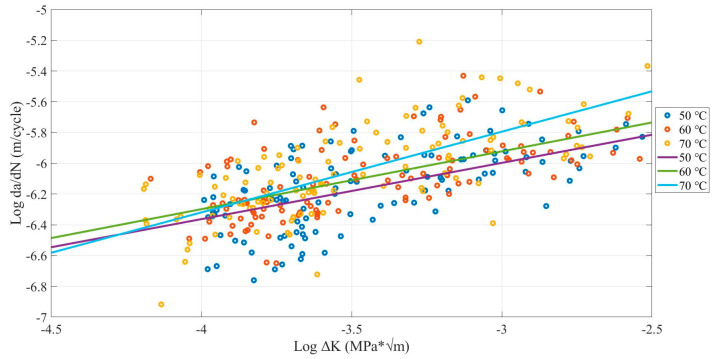
Comparison between the environmental temperatures influence on FCG rate of FDM ABS samples printed with (0.05-, 0.10-, and 0.15-mm layer thickness; 0.4-, 0.6-, and 0.8-mm nozzle sizes; and X, XY, and Y building orientations).

**Table 1 polymers-13-03737-t001:** Parameters of printed samples.

Building Orientation	Nozzle Size	Layer Thickness
X (0°)	0.4 mm	0.05 mm
XY (±45°)	0.6 mm	0.10 mm
Y (90°)	0.8 mm	0.15 mm

**Table 2 polymers-13-03737-t002:** Filament specifications, and mechanical and thermal properties of the Ultimaker^®^ ABS (RAL 3020) [33].

Filament Specifications and Properties	Value
Diameter	2.85 ± 0.10 mm
Tensile modulus	1681 MPa (ISO 527)
Tensile stress at yield	39 MPa (ISO 527)
Tensile stress at break	33.9 MPa (ISO 527)
Elongation at yield	3.5% (ISO 527)
Elongation at break	4.8% (ISO 527)
Melt mass flow rate (MFR)	41 g/10 min (ISO 1133)
Melting temperature	225–245 °C (ISO 294)
Glass transition temperature	97 °C (ISO 294)

**Table 3 polymers-13-03737-t003:** Standardized Value for different parameters.

Building Orientation	Nozzle Size	Layer Thickness	Temperature	Standardised Value
X (0°)	0.4 mm	0.05 mm	50 °C	−1
XY (±45°)	0.6 mm	0.10 mm	60 °C	0
Y (90°)	0.8 mm	0.15 mm	70 °C	1

**Table 4 polymers-13-03737-t004:** R-squared and RMSE for curve fitting of each parameter.

Parameters		R-Squared	RMSE
Building orientation	X (0°)	0.6251	0.1562
	XY (±45°)	0.6244	0.1752
	Y (90°)	0.5061	0.175
Nozzle size (mm)	0.4	0.4103	0.2054
	0.6	0.4235	0.2057
	0.8	0.3556	0.2253
Layer thickness (mm)	0.05	0.4517	0.247
	0.10	0.4178	0.1932
	0.15	0.3077	0.198
Temperature (°C)	50	0.3222	0.2143
	60	0.3862	0.1937
	70	0.4781	0.2281

**Table 5 polymers-13-03737-t005:** The mean FCG rate and number of cycles of different orientations regardless of other printing parameters.

Building Orientation	Mean FCG Rate (m/cycle)	Mean Number of Cycles until the Fracture
X	9.68 × 10^−7^	4343
XY	9.75 × 10^−7^	3912
Y	9.77 × 10^−7^	2282

**Table 6 polymers-13-03737-t006:** The mean FCG rate and number of cycles of different layer thicknesses regardless of other printing parameters.

Layer Thickness (mm)	Mean FCG Rate (m/cycle)	Mean Number of Cycles until the Fracture
0.05	1.04 × 10^−6^	3417
0.10	9.94 × 10^−7^	3523
0.15	9.19 × 10^−7^	3805

**Table 7 polymers-13-03737-t007:** The mean FCG rate and number of cycles of different nozzle sizes regardless of other printing parameters.

Nozzle Size (mm)	Mean FCG Rate (m/cycle)	Mean Number of Cycles until the Fracture
0.4	1.06 × 10^−6^	3224
0.6	1.02 × 10^−6^	3374
0.8	8.19 × 10^−7^	4147

**Table 8 polymers-13-03737-t008:** The mean FCG rate and number of cycles of different environmental temperatures regardless of other printing parameters.

Environmental Temperature (°C)	Mean FCG Rate (m/cycle)	Mean Number of Cycles until the Fracture
50	9.77 × 10^−7^	3604
60	9.78 × 10^−7^	3292
70	9.86 × 10^−7^	3122

**Table 9 polymers-13-03737-t009:** The Paris power law constant (C and m) for a part of configurations obtained from experimental results where (da/dN in m/cycle and Δ*K* in MPa*√m).

Building Orientation	T (°C)	Nozzle Size (mm)	Layer Thickness (mm)	Log C	C	m	R-Square (%)
X	50	0.4	0.05	−3.607	2.47 × 10^−4^	0.763	94%
X	50	0.4	0.10	−4.885	1.30 × 10^−5^	0.390	81%
X	50	0.4	0.15	−5.099	7.96 × 10^−6^	0.329	91%
X	50	0.6	0.05	−3.754	1.76 × 10^−4^	0.754	99%
XY	50	0.4	0.05	−2.078	8.36 × 10^−3^	1.131	98%
Y	50	0.4	0.05	−2.938	1.15 × 10^−3^	0.818	84%
Y	50	0.8	0.15	−4.660	2.19 × 10^−5^	0.418	98%
Y	60	0.8	0.15	−4.660	2.19 × 10^−5^	0.418	98%
Y	70	0.8	0.15	−3.408	3.91 × 10^−4^	0.731	99%

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
