# Peer review of "Modelling and Investigation of Crack Growth for 3D-Printed Acrylonitrile Butadiene Styrene (ABS) with Various Printing Parameters and Ambient Temperatures"

_polymers, 2021, doi:10.3390/polym13213737_

Round 1
Reviewer 1 Report
"Modelling and Investigation of Crack Growth for 3D-printed ABS with Various Printing Parameters and Ambient Temperatures" is an interesting work dealing with the investigation of FCG and SIF of 3D-printed ABS. Authors have shown the influence of layer thickness, temperature, and other parameters on the stresses and mechanical properties.
I have few minor remarks:
1) For each linear approximation and each parameter it would be essential to report R2 (on the graph) and layer show the error for the processed parameyters.
2) Check your citations. Sometimes you citing papers with strange style, e.g. Line34 [1][2][3][4]. It should be [1-4]. It appears in multiple places, please check through the paper.
Reviewer 2 Report
In this manuscript the authors investigate three essential parameters of FDM 3D printing (building orientation, layer thickness, and nozzle size) and their their influence on fatigue crack growth (FCG) rate under dynamic loads and the Paris power law constant C and m.
Moreover, the authors propose an analytical solution to determine the stress intensity factor (SIF) at the crack tip based on the measurement of structural dynamic response.
This is indeed an interesting work. Nevertheless some minor revision is needed in order to publish this work.
- The authors used a filament of 2.85±0.10mm diameter. Could they make a comment/estimation regarding a dia of 1.75±0.10mm? Thus their findings could be valid for all FDM 3D printers.
- I feel that section 3.1 "Calculation of SIF Range and FCG rate" should be supported by more references. Could the authors do that?
- I would liketo see real photographs (either by optical microscopy or even SEM) in Figures 1, 2 and 3. Could the authors present them along with the sketches their are showing?
- Indeed, there is not an ASTM standard for FDM polymers; The bending fatigue test of FDM polymers does not have a specific standard. There are other works in the literature where a simple parallelepiped is being used. The authors state that "..There is a 0.5 mm deep intial-seeded crack near the fixed end of the beam for all the samples to ensure that the maximum stress concentration took place at the same point..". I suggest the authors to present some references and compare other geometries in bending tests with their own.
- Some SEM pictures from the cracked specimens would be helpful. I hope the authors could provide them.
- A few typos and grammatical issues need to be resolved.
I suggest the authors to revised their manuscript following the above minor issues.
